# Feasibility of Cell-Free DNA Measurement from the Earlobe during Physiological Exercise Testing

**DOI:** 10.3390/diagnostics12061379

**Published:** 2022-06-02

**Authors:** Nils Haller, Aleksandar Tomaskovic, Thomas Stöggl, Perikles Simon, Elmo Neuberger

**Affiliations:** 1Department of Sports Medicine, Rehabilitation and Disease Prevention, Johannes Gutenberg University of Mainz, 55099 Mainz, Germany; nhaller@uni-mainz.de (N.H.); altomask@uni-mainz.de (A.T.); simonpe@uni-mainz.de (P.S.); 2Department of Sport and Exercise Science, University of Salzburg, 5400 Rif, Austria; thomas.stoeggl@plus.ac.at; 3Red Bull Athlete Performance Center, 5303 Thalgau, Austria

**Keywords:** cell-free DNA, exercise physiology, liquid biopsy, load monitoring, physiological exercise testing

## Abstract

Circulating, cell-free DNA (cfDNA) has been discussed as an upcoming blood-based biomarker in exercise physiology, reflecting important aspects of exercise load. cfDNA blood sampling has evolved from elaborate venous to efficient capillary sampling from the fingertips. In this study, we aimed to evaluate the principal feasibility of cfDNA blood sampling from the earlobe. Therefore, we obtained cfDNA concentrations from the fingertips, earlobe, and the antecubital vein during physiological exercise testing. Significantly higher concentrations were obtained from the earlobe compared to fingertip samples. All of the measurement methods showed good to excellent repeatability (ICCs of 0.85 to 0.93). In addition, the control experiments revealed that repeated sampling from the earlobe but not from the fingertips increased cfDNA at rest. In summary, cfDNA sampling is feasible for all sampling sources. However, at rest, cfDNA collected from the earlobe tend to increase over time in the absence of physical load, potentially limiting this sampling method.

## 1. Introduction

The existence of cell-free circulating DNA was first described in 1948 by Mandel and Métais [1]. These cell-unbound DNA fragments can be detected in healthy individuals in sizes ranging from less than 100 to more than 10,000 base pairs [2,3]. Over the years, it has been shown that cfDNA can be detected under various acute and chronic pathological conditions such as cancer [4,5], systemic lupus erythematosus [6], stroke [7], or sepsis [8]. As possible sources, passive cell death mechanisms such as apoptosis or necrosis have been discussed [9].

Recent studies have highlighted the importance of cfDNA in the field of exercise physiology [10,11,12]. Here, cfDNA is thought to be actively released from granulocytes through extracellular trap formation [13,14]. cfDNA was shown to increase during endurance exercise as a result of intensity and duration [15]. In addition, cfDNA was found to have a similar intensity-dependent release during strength training [16]. Haller and colleagues [17] have, moreover, observed a correlation with total distance covered in soccer, further highlighting the potential of cfDNA as an objective variable in objective monitoring and quantification of exercise load.

In the early years, cfDNA was mainly quantified from elaborate venous blood samples. However, measurement methods have evolved to efficient low-volume capillary blood sampling from the fingertip, with Breitbach and colleagues [18] showing comparable results between the measurement methods. From a practical point of view, athletes may perceive repeated sampling from the finger uncomfortable, with the risk of inflammation at the sampling point. This can be exacerbated in sports that are primarily performed with the hand, such as handball or basketball. Earlobe sampling could be another step forward, as this sampling is usually perceived as comfortable and athletes are already accustomed to it from regular lactate sampling [19]. However, whether capillary blood from the earlobe contains comparable cfDNA concentrations to capillary blood from the finger at rest or during exercise has never been the subject of investigation.

Therefore, in this study, we aimed to determine whether cfDNA sampling from the earlobe is feasible in principle, exhibits similar concentrations and kinetics during exercise as concentrations from the fingertip and from the vein, and correlates acceptably with each other in a controlled test-retest setting.

## 2. Materials and Methods

### 2.1. Ethical Approval

All of the procedures were approved by the Ethics Committee of the State of Rhineland-Palatinate, Germany and met the standards of the Declaration of Helsinki of the World Medical Association. All of the participants were informed about the experimental design and the aim of the study and gave their written consent to participate.

### 2.2. Subjects, Setting

Figure 1 outlines the study design. All of the participants started the experiment with a medical screening prior to the tests. Five participants underwent an incremental running test (physiological exercise testing) with 3 of those 5 participants completing a test-retest (resulting in *n* = 8 tests in total).

Physiological exercise testing started at a speed of 6 km/h for 3 min and increased by 2 km/h after each step until voluntary exhaustion. Pause time between the steps was 30–45 s for blood sampling. VO_2,_ VO_2__max_, heart rate (HR; via a 12-channel ECG) and lactate concentrations including lactate threshold (LT, using the baseline +1.5 mmol/L model [20]) were determined in all test settings for a descriptive assessment of participant performance. Breath-by-breath measurement technique was used to determine oxygen uptake.

Capillary blood from the earlobe (lactate and cfDNA) and from fingertips (cfDNA) was taken prior to the tests, after every step, as well as post-exercise (directly post-exercise, +5-, and +15-min post-exercise). At the earlobe, lactate was collected first with subsequent collection of cfDNA via the same puncture site. Each time blood was drawn from the fingertip, a new prick was made on a different finger. Venous blood (cfDNA) was taken before and directly after the incremental test. The rate of perceived exertion (RPE, Borg-Scale from 6–20, [21]) was requested from participants after each step and after completion of the test.

To determine whether the increase in cfDNA concentrations during exercise in any of the collection methods might be affected by repeated blood collection, a control experiment was performed. At rest, in each setting with 2 participants cfDNA was collected, (i) from the fingertip nine consecutive times, from the earlobe nine consecutive times (ii) without hyperemizing ointment, and (iii) with hyperemizing ointment.

### 2.3. Quantification of cfDNA and Lactate

Approximately 15–20 μL of capillary blood from the earlobe and the fingertip was collected in Microvette^®^ CB 300 K2E (Sarstedt, Nümbrecht, Germany) and centrifuged at 1600× *g* for 10 min. The remaining plasma was then transferred and stored at −20 °C. In addition, approximately 7.5 mL of venous blood was collected from the antecubital vein using S-Monovette^®^ 7.5 mL, K3 EDTA (Sarstedt, Nümbrecht, Germany).

Concentrations of cfDNA were quantified by analysis of unpurified plasma by quantitative real-time PCR (qPCR) as described elsewhere [22]. Briefly, plasma was diluted (1:10 in H_2_O) and was used as template for qPCR. Amplification was based on primers (5′-TGCCGCAATAAACATACGTG-3′ and 5′-GACCCAGCCATCCCATTAC-3′) targeting a 90 bp fragment of human long nuclear elements (LINEs) of the repetitive L1PA2 family. A CFX384 Bio-Rad (Bio-Rad, Munich, Germany) cycler was used with the following cycling conditions: 2 min 98 °C heat activation, followed by a two-step protocol with 10 s 95 °C denaturation, and 10 s 64 °C annealing, for 35 cycles with subsequent melting curve from 70 to 95 °C with 0.5 °C increments for 10 s. Each sample was measured as a technical triplicate with 5 µL final volume per well. The reaction mix contained 0.66 µL of 1:10 diluted plasma, 0.33 µL primer mix (140 nm final concentration of each primer) and 4 µL of qPCR mix with 0.6 U Velocity Polymerase (Bioline, London, UK), 1.2 × Hifi Buffer (Bioline, London, UK), 0.1 × SYBR Green (Sigma, St. Louis, MO, USA), 0.3 mM dNTPs (Bioline, London, UK). The concentrations of the qPCR mix reflect the final concentrations per 5 µL reaction.

Ahead of the measurements, the linearity, limit of quantification, and limit of detection of the assay were established [22]. In addition, a set of two reference samples were pre-validated and were included in each run. To calculate the amount of DNA, the predefined intercept and slope of the assay were used to calculate the number of molecules in a 5 µL qPCR reaction using the following formula: 10^(Cq-intercept)/slope^. To calculate the concentration of DNA in the plasma samples the following formula was applied: ng/mL = pg/µL = 10^(Cq-intercept)/slope^/5 µL/dilution factor of the sample × 3.23 pg/3416. A division by 5 µL is required to calculate the genome equivalents (GE) per µL qPCR reaction. The final dilution factor of the sample equals 1:75, respecting the 1:10 dilution of plasma, of which 0.66 µL were included in a 5 µL reaction. The resulting GE were multiplied with the weight of a haploid human genome and divided by 3416, the number of hits of the amplified target in the human genome [22].

A total of 20 μL of capillary blood was collected and analyzed with the Biosen 5130 (EKF Diagnostics, Magdeburg, Germany).

### 2.4. Statistical Analysis

The qPCR data were acquired using CFX Manager 3.0 (Bio-Rad, Munich, Germany). Microsoft^®^ Excel 2018 (Microsoft Corp, Redmond, WA, USA) was used for data collection. We considered *p*-values < 0.05 to be statistically significant and performed statistical analysis with JMP 13 (SAS, Cary, NC, USA), SPSS 23 (IBM, Chicago, IL, USA) and R (version 4.1.2, R Foundation for Statistical Computing, Vienna, Austria). Figures were created using the ggplot 2 package (version 3.3.5).

The cfDNA concentrations were log-transformed to achieve a normal distribution of concentrations. A linear mixed effects model (“lmer” and “lmerTest:anova” in library “lmerTest”, v3.1-3) was used to test for differences between fingertip and earlobe samples, and to test if log10 cfDNA increased during exercise, using “sample source” and “time point” as fixed factors and with the subject as random effect.

In a separate analysis, we tested the correlation between physiological variables and cfDNA concentrations (fingertip, earlobe, venous) using the nonparametric Spearman rank test. Accordingly, we used Bonferroni-corrected *p*-values (meaning we multiplied the *p*-values by the number of total comparisons, *n* = 21).

Repeatability of fingertip, earlobe, and venous cfDNA sampling was tested by calculating intraclass correlation coefficients (ICC). According to Koo&Li [23], ICC values less than 0.5 indicate poor reliability, between 0.5 and 0.75 indicate moderate reliability, between 0.75 and 0.9 indicate good reliability, and above 0.90 indicate excellent reliability. ICC estimates and their 95% confident intervals (CI) were calculated based on a mean-rating, absolute-agreement, and the 2-way mixed-effects model.

To evaluate the agreement of cfDNA measurement in capillary blood samples from the fingertip and earlobe, Bland & Altman analysis [24] with d = mean of cfDNA differences (earlobe cfDNA–fingertip cfDNA) and limits of agreement (LoA, d ± 1.96 × sd, sd = standard deviation of the cfdna differences) was performed.

## 3. Results

### 3.1. Participant Characteristics

Table 1 lists participant characteristics. Two participants were unable to complete a retest. Therefore, the first column contains all 8 completed tests, and consequently 3 subjects are included twice in this column due to the retests. Therefore, 2 additional columns were created, 1 containing the data from test 1 with all 5 participants and 1 column containing the data from the 3 retests only.

### 3.2. cfDNA during Exercise

Compared to the fingertip, earlobe samples showed significantly higher overall concentrations of log10 cfDNA (Figure 2a), (F_1,130_._28_ = 13.76, *p* < 0.001), and increased over incremental running time (F_10,130_._60_ = 110.49, *p* < 0.001) with no significant interaction between sample source and time point (F_10,130_._28_ = 1.58, *p* = 0.19). Post-hoc T-test analysis revealed significantly higher earlobe cfDNA concentrations at +5 min, and +15 min compared to fingertip. Concentrations obtained at speeds 18 and 20 km/h were excluded from this analysis since only one subject performed these steps in both tests. Regarding post-exercise concentrations, cfDNA concentrations were significantly higher in earlobe concentrations compared to venous samples (*p* = 0.04, Figure 2b).

With respect to repeatability, the ICCs for fingertip cfDNA from between test and retest (29 comparisons in total) were 0.93 (95% CI: 0.83–0.97). The earlobe cfDNA ICC (29 comparisons in total) was calculated to be 0.85 (95% CI: 0.67–0.93), whereas the ICC from the venous cfDNA measurement was calculated to be 0.93 (95% CI: 0.56–0.99), but with a total of only 6 comparisons.

Evaluating the agreement of earlobe and fingertip (Figure 3) cfDNA measurement in capillary blood samples, higher cfDNA levels were observed from earlobe, with a mean log10 difference of 0.09, with LoA of –0.29 and 0.47 (78 comparisons in total).

### 3.3. cfDNA Kinetics at Rest

Figure 4 outlines a descriptive view of 6 cases of cfDNA measurements obtained at rest (control experiment) with no exercise load induced. Due to the low sample size, a descriptive view is provided. Apparently, fingertip concentrations remain constant at baseline, with one outlier in case 2 (min 15). The sample was re-analyzed; however, a similar concentration was obtained. The earlobe concentrations appear to increase in all cases with higher increases after usage of hyperemizing ointment at the sampling site.

### 3.4. Correlations between Variables

Figure 5 outlines correlations between physiological variables and cfDNA from different sample sources. With two exceptions, variables significantly correlated with each other.

## 4. Discussion

In this study, we have demonstrated for the first time that cfDNA blood sampling from the earlobe is possible in principle. We observed quite similar kinetics during physiological exercise testing compared to venous and capillary cfDNA samples from the fingertip. Interestingly, overall concentrations from the earlobe were significantly higher compared with concentrations from the fingertip. Differences were more pronounced at higher exercise intensity and immediately after exercise, which could be associated with wound healing process. Despite this, all of the sampling methods provided good-to-excellent repeatability. In a control experiment, cfDNA from the earlobe, which was collected from a single punctation side, showed a tendency to increase at rest even without the induction of physical load with fingertip samples remaining stable over time. Notably, for sample collection from the finger, a new punctation is required in order to avoid such biological bias of wound healing.

The kinetics of cfDNA during incremental running are consistent with previous studies showing a linear increase followed by a rapid decrease post-exercise [18,25]. Interestingly, both fingertip and earlobe cfDNA showed quite similar kinetics with significantly higher concentrations at the earlobe. Regulated cell death (such as apoptosis, necrosis, autophagy, and NETosis) as well as the release from dying cells and accidental cell death (rapid death in response to destructive mechanical, physical, or chemical perturbations) may determine the presence of cfDNA in blood [26]. Tug et al. [27] demonstrated that cfDNA is primarily released by blood cells from the haemopoietic lineage and Neuberger et al. [14] described neutrophil granulocytes as the main source of cfDNA during exercise.

In this context, several studies have compared hematological variables of capillary blood from the fingertip with venous blood samples and found a significantly higher proportion of white blood cells in capillary blood [28,29,30]. Therefore, it is not unexpected that the concentrations at the different sampling sites are not completely similar. However, the increased concentrations due to repeated pressure at rest and during exercise at the earlobe are somewhat surprising. Yang et al. [28] observed changes in blood composition due to repeated sampling from the fingertip, for instance in leukocytes which, however, tend to decrease. This finding, nevertheless, highlights the fact that repeated pressure on the puncture site can alter the composition of the blood.

Increased concentrations measured at the earlobe at rest and during exercise could be due repeated pressure (mechanical perturbation) on an already existing wound leading to an immediate thrombocyte and granulocyte aggregation (immunothrombosis) after skin punction [28,31] and an inflammatory response at the sampling site that starts within seconds [32,33]. It is already known from surgery that there is a positive correlation between the extent of surgical trauma and cfDNA levels [34,35]. Yu et al. [32] found that hemodynamic forces trigger NETosis, which could provide an explanation for the increased concentrations at the earlobe [12]. In addition, compression of the sampling site on both earlobe and fingertip and/or repeated squeezing may cause hemolysis and mixing of interstitial and intracellular fluid with the blood sample [36]. However, in addition to the destruction of red blood cells (in vitro hemolysis), white blood cells may also be physically damaged during blood collection. Distinct from erythrocytes, white blood cells contain nuclear DNA, and their lysis during in vitro hemolysis result in cellular DNA being released into the blood [37].

This potential biological bias from increased concentrations could limit the practicality of repeated earlobe sampling although, similarly to fingertip and venous cfDNA sampling, it demonstrated strong correlations with physiologically meaningful variables (Figure 5). HR, VO_2_, Borg-scale and lactate concentrations are regularly used by practitioners and researchers to evaluate and monitor exercise intensity [38,39]. Significant correlations of cfDNA (fingertip, earlobe, venous) with those variables are consistent with the previous findings of Haller et al. [15] and Breitbach et al. [18]. These strong correlations underline the potential application of cfDNA as a suitable, and reliable marker in the context of physiological exercise testing. On the other hand, the correlations of lactate, which is often reported as the gold standard for monitoring intensity in physiological exercise testing, with performance variables (VO_2_, HR, Borg), are comparable or slightly worse. This is most likely due to the fact that lactate concentration in incremental exercise testing typically decrease at low intensities and usually exhibits exponential kinetics after the initial steps [38,40], whereas cfDNA usually shows a rather linear progression [15,18].

## 5. Conclusions

Both earlobe and fingertip sampling are feasible for cfDNA sampling in principle. Both methods were well-correlated and aligned with other performance variables. However, overall and particularly at some steps during the incremental test, concentrations significantly differed, with higher concentrations occurring at earlobe. Repeated pressure on the earlobe could have triggered increased inflammation leading to increased cfDNA concentrations. Thus, these results from the earlobe should be viewed with caution. To overcome this obstacle, cfDNA samples could be collected from the earlobe before and after exercise at different earlobe, or otherwise, the fingertip could be used, as in previous studies.

## Figures and Tables

**Figure 1 diagnostics-12-01379-f001:**
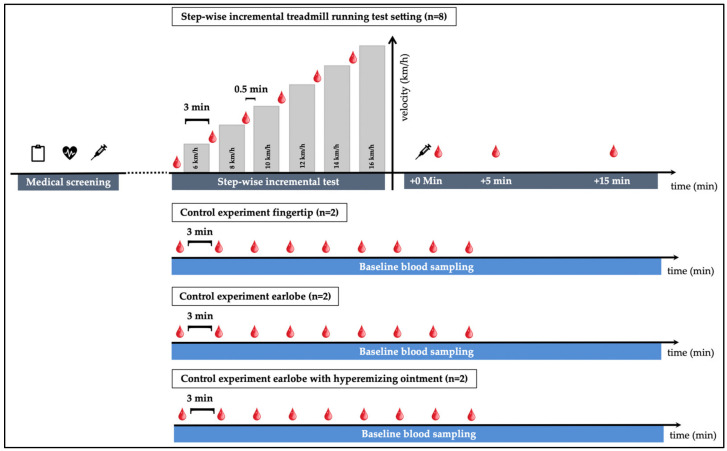
Study outline: 5 participants, three of whom performed a retest, performed a total of 8 incremental exercise tests. Tests started at 6 km/h with increases of 2 km/h and a duration of 3 min per step. Pause time between steps was roughly 0.5 min. A total of 3 additional control experiments were performed: 9 repeated blood samplings at rest with 3 min in between from (i) alternating fingertips (*n* = 2), (ii) from the earlobe without hyperemizing ointment, (iii) from the earlobe with an hyperemizing ointment (*n* = 2).

**Figure 2 diagnostics-12-01379-f002:**
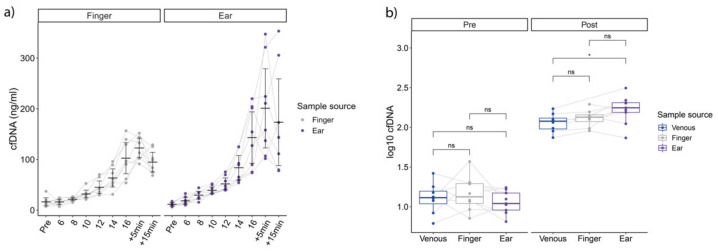
(**a**) shows cfDNA concentrations on all steps from the fingertip (left) and the earlobe. (**b**) outlines pre to post concentrations. Significant differences determined via *T*-test (*p* < 0.05) in ear compared to venous concentrations, without post-hoc correction.

**Figure 3 diagnostics-12-01379-f003:**
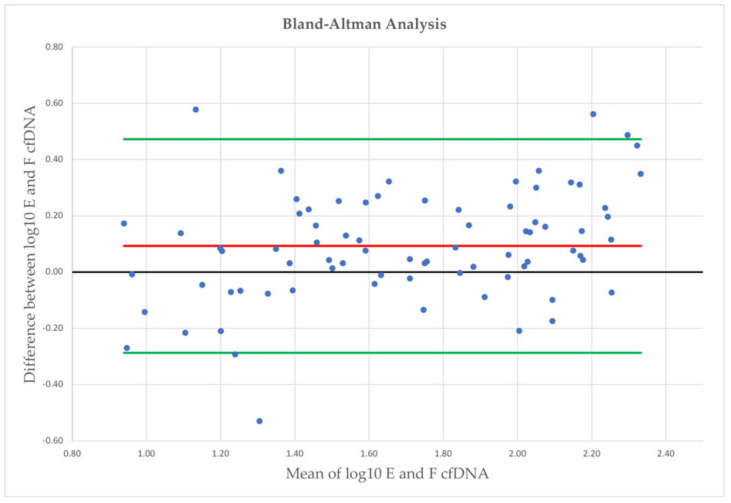
Bland-Altman plot analysis with mean log10 concentrations on the *x*-axis and differences between both measurements on the *y*-axis. E = earlobe; F = fingertip; red line = mean log10 difference; green lines = limits of agreement.

**Figure 4 diagnostics-12-01379-f004:**
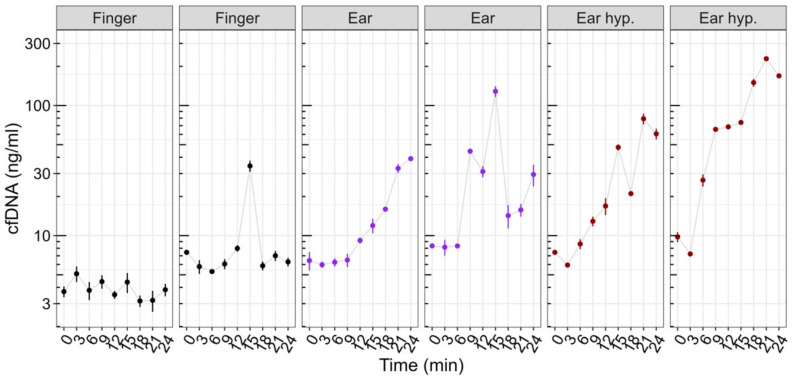
Control experiment at rest using fingertip (black) and earlobe without (purple) and with hyperemizing ointment (brown). Fingertip concentrations with one outlier (case 2, min 15). Earlobe sampling with several outliers and a tendency to increase over time with and without the use of hyperemizing ointment at the sampling site. The points and error bars represent the mean and SD for the technical triplicate determination of cfDNA concentration.

**Figure 5 diagnostics-12-01379-f005:**
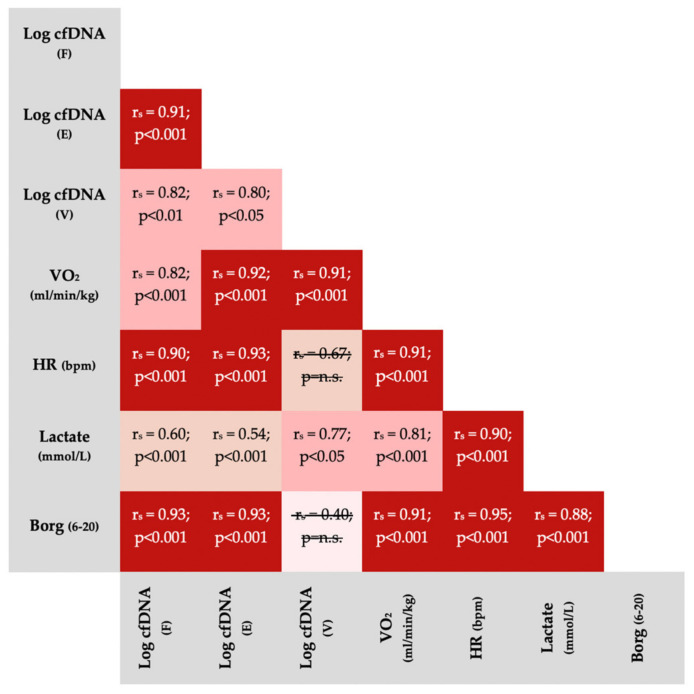
Spearman Correlations (r_s_) between variables obtained during incremental running tests (all participants included). All *p*-values were Bonferroni corrected for 21 comparisons. Non-significant correlations were crossed out. The non-significant correlation between venous cfDNA and Borg-scale is based on a small sample size (*n* = 7) because only post-exercise Borg scores, usually reported as 19 or 20, could be compared with venous cfDNA. VO_2_ = oxygen consumption; HR = heart rate; bpm = beats per minute; E = earlobe; F = fingertip; V = venous.

**Table 1 diagnostics-12-01379-t001:** Participant characteristics.

Parameter	All Tests (*n* = 8)	Test (*n* = 5)	Retest (*n* = 3)	Control Experiment (*n* = 6)
**Anthropometric variables**				
Age (yrs.)	28 (2)	28 (2)	28 (1)	31 (13.3)
Body weight (kg)	78.1 (4.3)	80.5 (1.9)	76.7 (4.7)	84.2 (6.0)
Body height (cm)	185.6 (3.1)	186.1 (3.2)	185.3 (2.9)	181.3 (10.9)
Body mass index (kg/m^2^)	23.5 (2.4)	23.7	23.6 (3.1)	26 (4.2)
**Physiological variables**				
lactate threshold (km/h)	12.3 (1.1)	12.3 (1.2)	12.5 (0.8)	
lactate at baseline (mmol/L)	1.2 (0.4)	1.1 (0.5)	1.4 (0.4)	
lactate maximum in incremental test (mmol/L)	11.3 (2.6)	10.8 (3.1)	12.1 (1.6)	
heart rate at start of incremental test (beats/min)	72 (11)	71 (11)	75 (14)	
heart rate maximum in incremental test (beats/min)	188 (6)	188 (6)	189 (8)	
cfDNA venous before incremental test (ng/mL)	12.2 (3.9)	10.8 (4.2)	14.2 (3.2)	
cfDNA venous after incremental test (ng/mL) *	110.6 (25)	96.7 (21.4)	119.5 (5.9)	
cfDNA venous mean fold-change (post to pre)	9.6 (2.1)	9.8 (2.8)	9.7 (0.9)	
cfDNA finger before incremental test (ng/mL)	16.4 (9.5)	14.1 (5.9)	20.3 (14.6)	
cfDNA finger after incremental test (ng/mL)	134.2 (33.2)	129. 6 (41.9)	141.8 (13.7)	
cfDNA finger mean fold-change (post to pre)	10.1 (4.7)	10.5 (5.1)	9.3 (4.9)	
cfDNA earlobe before incremental test (ng/mL)	11.5 (3.9)	13.2 (3.7)	8.6 (2.2)	
cfDNA earlobe after incremental test (ng/mL)	163.2 (49.8)	167.3 (56.5)	156.5 (46.8)	
cfDNA earlobe mean fold change (post to pre)	15.3 (6.3)	12.8 (4.1)	19.4 (8)	
Borg scale at exhaustion	19.6 (0.6)	19.6 (0.5)	19.6 (0.6)	
Maximal oxygen consumption (VO_2max_) **	51.8 (6.5)	50.7 (7.4)	54.5 (3.7)	

* No venous samples of one participant at “test” (“all tests” *n* = 7 and “test” *n* = 4) ** No VO_2max_ assessment of one participant at “Retest” (“all tests” *n* = 7 and “Retest” *n* = 2).

## Data Availability

The data presented in this study are available on reasonable request from the corresponding author.

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
