# Peer review of "Feasibility of Cell-Free DNA Measurement from the Earlobe during Physiological Exercise Testing"

_diagnostics, 2022, doi:10.3390/diagnostics12061379_

Round 1

Reviewer 1 Report

This manuscript compared and analyzed the cfDNA concentration in blood samples from fingertips, earlobe, and antecubital vein during physiological. This study concludes that the cfDNA that was obtained from earlobe tends to increase along with multiple sampling. The study is overall well designed, and the quality of the article is satisfactory. I suggest this manuscript can be accepted with minor revision.

Addition comments:

  1. Figure 1 is somewhat confusing. It would be nice if the authors can add units right behind each number in the bar graph.
  2. The data in table 1 can be visualized with bar graphs for readers’ convenience.
  3. In the article, the abbreviation “E”, “F”, “V” were used to substitute earlobe, fingertips, and veinous samples. However, the use of these abbreviations may make the article harder to read.
  4. Please provide the PCR protocol for the quantitative measurement of cfDNA. Was any standard ssDNA sample used to calibrate the qPCR assay? Was a DNA extraction step involved before the qPCR assay?
  5. The 90bp fragment of LINEs was used as the target of the qPCR test. However, in all the graphs, cfDNA (ng/ml) was labeled on Y-axis. Does it mean the mass concentration of all cfDNA in the blood sample, or only the 90bp fragment? 
  6. Why did the authors choose not to collect the veinous blood sample during the test?
  7. In figure 4, the data seems inconsistent along the time. It seems that only one qPCR test was performed on each blood sample. Since a sufficient volume of blood sample (>15μL) was collected, each qPCR test can be replicated at least 3 times to overcome the bias and errors.
  8. Figure 5 showed that the cfDNA from the fingertips and the vein were not correlated. Could the authors discuss the possible reason?
  9. The authors stated that the increase of cfDNA in earlobe blood was caused by repeating sampling at the same punctation side. It would be better if the authors can add another data set of fingertip blood that is collected at the same punctation site over time (at rest)?

Reviewer 2 Report

Review Report

The manuscript describes the feasibility of using cell-free DNA collected from the earlobe as a marker during physiological exercise testing.

The authors used three sources of cell-free DNA: venous blood, capillary blood from the fingertip, and capillary blood from the earlobe. They wanted to test whether the amount of cell-free DNA collected from the earlobe (least invasive) is similar to venous blood or fingertip collected samples. cfDNA concentrations were determined by real-time PCR. Several statistical approaches were used to assess the data collected.

The authors show that cfDNA determination is possible from the earlobe. However, repeated sampling from the earlobe, also in the absence of physiological load, tends to increase the amount of cfDNA in each consecutive sample.

Generally, the manuscript is informative, reads well, and is logically structured. I would have some recommendations, suggestions:

Minor:

1.)    Line 56: Ethical approval

Additional information regarding the ethical approval (number of the approval, code etc.) is needed as also stated in the Instructions for Authors.

2.)    Line 93: Quantification of cfDNA

Since real-time PCR was used to quantify cfDNA there is very little information here that would enable someone to repeat this experiment. The authors need to significantly improve this section with additional information about PCR volumes, PCR cycling conditions etc.

 I could also not find any data as to how the authors arrived at the ng/mL cfDNA amount which they used in table 1 from real-time PCR data?

 3.)    Line 163: Figure 3

There is no mention of Figure 3 in the text.
